# The Effect of Antioxidants on Photoreactivity and Phototoxic Potential of RPE Melanolipofuscin Granules from Human Donors of Different Age

**DOI:** 10.3390/antiox9111044

**Published:** 2020-10-26

**Authors:** Magdalena M. Olchawa, Grzegorz M. Szewczyk, Andrzej C. Zadlo, Michal W. Sarna, Dawid Wnuk, Tadeusz J. Sarna

**Affiliations:** 1Department of Biophysics, Faculty of Biochemistry, Biophysics and Biotechnology, Jagiellonian University, 30-387 Kraków, Poland; magdalena.olchawa@uj.edu.pl (M.M.O.); grzegorz.szewczyk@uj.edu.pl (G.M.S.); andrzej.zadlo@uj.edu.pl (A.C.Z.); michal.sarna@uj.edu.pl (M.W.S.); 2Department of Cell Biology, Faculty of Biochemistry, Biophysics and Biotechnology, Jagiellonian University, 30-387 Kraków, Poland; dawid.wnuk@uj.edu.pl

**Keywords:** melanolipofuscin, RPE cells, phagocytic activity, cell cytoskeleton, photic stress, singlet oxygen, superoxide anion, zeaxanthin, vitamin E

## Abstract

One of the most prominent age-related changes of retinal pigment epithelium (RPE) is the accumulation of melanolipofuscin granules, which could contribute to oxidative stress in the retina. The purpose of this study was to determine the ability of melanolipofuscin granules from younger and older donors to photogenerate reactive oxygen species, and to examine if natural antioxidants could modify the phototoxic potential of this age pigment. Electron paramagnetic resonance (EPR) oximetry, EPR-spin trapping, and time-resolved detection of near-infrared phosphorescence were employed for measuring photogeneration of superoxide anion and singlet oxygen by melanolipofuscin isolated from younger and older human donors. Phototoxicity mediated by internalized melanolipofuscin granules with and without supplementation with zeaxanthin and α-tocopherol was analyzed in ARPE-19 cells by determining cell survival, oxidation of cellular proteins, organization of the cell cytoskeleton, and the cell specific phagocytic activity. Supplementation with antioxidants reduced aerobic photoreactivity and phototoxicity of melanolipofuscin granules. The effect was particularly noticeable for melanolipofuscin mediated inhibition of the cell phagocytic activity. Antioxidants decreased the extent of melanolipofuscin-dependent oxidation of cellular proteins and disruption of the cell cytoskeleton. Although melanolipofuscin might be involved in chronic phototoxicity of the aging RPE, natural antioxidants could partially ameliorate these harmful effects.

## 1. Introduction

Retinal pigment epithelium (RPE) is the outermost part of the eye retina consisting of a single layer of cells adjacent to photoreceptors cells. RPE is not only an important component of the blood–retina barrier, it also provides metabolic support for the entire retina [1,2]. One of the key biological functions of RPE is the efficient phagocytosis of membranes of photoreceptor outer segments (POS) that are periodically shed off [3,4]. RPE of a normal human eye contains melanin granules, viewed as a photoprotective pigment and natural antioxidant [5,6]. A mechanical role of melanosomes in the RPE has also been postulated [7]. Melanin in the human RPE is formed early during fetal development and, afterwards, shows little or no metabolic turnover. Being post-mitotic, RPE cells undergo significant morphological changes with aging; their size increases [8,9], their melanin content decreases, and they accumulate two distinct age pigments—lipofuscin and melanolipofuscin, which could contribute to oxidative stress of the cells [8,9]. Oxidative stress has been postulated to play an important role in neurodegenerative diseases, such as age-related macular degeneration (AMD) which is the predominant cause of blindness in people over 65 in developed Western countries [10]. While the photoreactivity and phototoxicity of lipofuscin have been firmly established [11,12,13,14,15,16], the photochemical properties of melanolipofuscin granules and their phototoxic potential remain only partially known [17,18,19]. It is worth emphasizing that although the photoreactivity of lipofuscin granules might be higher than that of melanolipofuscin granules, the age-dependent accumulation of the latter pigment more closely reflects the onset of AMD. A recent study of human retinal pigment epithelium, using serial block face scanning electron microscopy, has demonstrated that while in the 32-year-old donor, the major fluorescent pigment is lipofuscin, in the 76-year-old donor melanolipofuscin becomes the predominant pigment [20]. Therefore, melanolipofuscin, rather than lipofuscin, might be the key age pigment of the human RPE responsible for photochemical reactions mediating chronic oxidative stress in the retina, which could lead to degenerative processes. 

The macular pigments zeaxanthin and lutein, due to their photophysical and antioxidant properties, are believed to protect the retina from photoinduced oxidative stress, and to prevent age-related macular degeneration (AMD) [21,22,23,24]. The AREDS study demonstrated that supplementation with carotenoids, zinc, vitamin C, and vitamin E reduced the 5-year risk of advanced AMD by 25% [25,26]. Higher intake of bioavailable lutein/zeaxanthin was found to be associated with a long-term reduced risk of advanced AMD [27]. In several model systems it was also demonstrated that antioxidant protection increased synergistically when combination of carotenoids and vitamin E was used [28,29,30]. Importantly, carotenoids are among the most efficient quenchers of singlet oxygen, while vitamin E is an efficient scavenger of peroxyl radicals [31,32,33,34]. Synergistic protection by zeaxanthin and vitamin E against photic stress in ARPE-19 cells, mediated by photosensitizing dyes [35], lipofuscin granules [13], or melanosomes [36], was shown in our previous studies. 

The main aim of this study was to determine the photoreactivity and phototoxicity of melanolipofuscin isolated from RPEs of human donors of different age, and to examine whether natural antioxidants could modulate the ability of the age pigment to photogenerate reactive oxygen species and disturb specific phagocytosis of RPE cells, their most important biological function.

## 2. Materials and Methods 

### 2.1. Materials

Minimum essential medium (MEM), trypsin, streptomycin, penicillin, gentamicin, amphotericin B, bovine serum albumin (BSA), chelating resins Chelex100, dialysis bags, catalase, diethylenetriaminepentaacetic acid (DTPA), 3-(4,5-dimethylthiazol-2-yl)-2,5-diphenyltetrazolium bromide (MTT), deuterium oxide, 5,5-dimethyl-1-pyrolineN-oxide (DMPO), and α-tocopherol were purchased from Sigma-Aldrich (Steinheim, Germany or St. Louis, MO, USA). Hank’s balanced salt solution and fetal bovine serum (FBS) was from Gibco-Invitrogen (Carlsbad, CA, USA or Auckland, New Zealand). *n*-Dodecyl-β-D-maltopyranoside (DDM) was purchased from Anatrace Products LLC (Maumee, OH, USA). The following chemicals were obtained from Polish Chemical reagents (POCH), (Gliwice, Poland): disodium hydrogen phosphate dodecahydrate (Na_2_HPO_4_ × 12H_2_O), potassium dihydrogen phosphate (KH_2_PO_4_), tris (hydroxymethyl) aminomethane, hydrochloric acid (HCl), sucrose, magnesium chloride hexahydrate (MgCl_2_ × 6H_2_O), potassium chloride (KCl), sodium chloride (NaCl), calcium chloride (CaCl_2_), ethylen diaminetetraacetic acid disodium salt (EDTA-Na_2_), sodium hydrogen carbonate (NaHCO_3_), ethanol, methanol and dimethyl sulfoxide (DMSO). 4-Protio-3-carbamoyl-2,2,5,5-tetraperdeuteromethyl-3-pyrroline-1-yloxy (mHCTPO) was a generous gift from Professor Howard J. Halpern (University of Chicago, Chicago, IL, USA). Zeaxanthin was a gift from F. Hoffmann-La Roche Ltd. (Basel, Switzerland). Fluorescein-5-isothiocyanate (FITC) was purchased from Molecular Probes (Eugene, OR, USA). Coumarin boronic acid (CBA) probe was a gift from the Institute of Applied Radiation Chemistry, Technical University (Lodz, Poland). All chemicals were reagent grade or better and used as supplied.

### 2.2. Isolation of Melanolipofuscin Granules (MLF) and Enrichment of Pigments Granules with Zeaxanthin and α-Tocopherol

RPE cells were scraped from eyecups of human donors no later than 24 h post-mortem by the employees of Niepubliczny Zaklad Opieki Zdrowotnej FRK Homograft Sp. z o. o, Zabrze, Poland, and frozen samples of RPE cells in PBS were made available to us. All experimental procedures were approved by the Bioethics Committee at the Jagiellonian University. Melanolipofuscin granules were isolated separately from the RPE cells of human donors pooled into two age groups; (a) 18–29 years old (RPE cells from 35 eyes) and (b) 50–59 years old (RPE cells from 35 eyes), according to the method described previously [13,36]. Briefly, thawed RPE cells were homogenized in PBS containing 0.1 mM EDTA and the obtained suspension was subsequently centrifuged four times (a. 80 g/7 min; b. 80 g/7 min; c. 5242 g/10 min; d. 5242 g/10 min) to remove large debris and collect the pellets. The obtained pellets were overlayed on a top of discontinuous sucrose density gradients (2.0, 1.8, 1.6, 1.55, 1.5, 1.4, 1.2, and 1.0 M) and centrifuged at 141,000× *g* for 1 h at 4 °C. Melanolipofuscin fraction identified as orange-brown bands at 1.6/1.8 M and 1.8/2.0 M interfaces was collected, washed three times, and suspended in phosphate-buffered saline (PBS). Concentration of pigment granules was determined by counting in a hemocytometer and de-aerated with argon suspensions of granules were stored in liquid nitrogen until further use.

To generate antioxidant preloaded granules for in vitro experiments, fresh aliquots of 5 mM zeaxanthin (ZEA) in tetrahydrofuran and 40mM α-tocopherol (TOC) in ethanol were quickly added to culture medium with antibiotics (penicillin, streptomycin, amphotericin, and gentamycin). Selected number of MLF granules from younger or from older donors were incubated in culture medium with a mixture of ZEA and TOC (at final concentrations 10 and 100 µM, respectively) for 24 h at 4 °C [13,36]. After incubation with antioxidants, MLF granules were centrifuged (5242 g/15 min), suspended in fresh medium, and added to cells.

For photoreactivity experiments in model systems and for detection of singlet oxygen, suspensions of MLF (1 × 10^8^ granules/mL or 1 × 10^9^ granules/mL) was enriched with antioxidants by incubation of the granules in PBS with ZEA and TOC (at final concentrations 10 and 100 µM, respectively) for 24 h at 4 °C, as described previously [13,36]. After incubation with antioxidants, MLF granules were centrifuged (5242 g/15 min), suspended in fresh PBS, and used in planned experiments.

### 2.3. Cell Culture Conditions and MLF Particle Administration

All experiments were performed on human retinal pigment epithelium (RPE) cell line ARPE-19 (American Type Culture Collection, Rockville, MD, USA). The cultures were plated in 96-well plates, 48-well plates, 6-well plates, or 25 cm^2^ flasks at a density 10^5^ cells/cm^2^ and propagated in Minimal Essential Medium (MEM) supplemented with 10% fetal bovine serum (FBS) and antibiotics (penicillin and streptomycin). Twenty-four hours after plating, cultures were loaded with MLF granules from donors of different age or antioxidants-enriched granules to ensure their phagocytic uptake during three subsequent feedings with 6 × 10^6^ granules/mL over every 72 h according to previously reported feeding procedure [13,36].

Twenty-four hours after the third feeding melanolipofuscin content in ARPE-19 cells was determined by electron paramagnetic resonance (EPR) spectroscopy at 77 K, employing Bruker EMX–AA EPR spectrometer, operating at X-band and with 100-kHz field modulation (Bruker BioSpin, Rheinstetten, Germany), as described previously [36,37,38]. Briefly, integrated EPR signal intensities of ARPE-19 cells preloaded with MLF granules were compared under the same experimental conditions (center field corresponding to g-value of approximately 2.00, scan range 7 mT, microwave power 0.0324 mW, and modulation amplitude 0.305 mT) with those of known number of isolated MLF granules and selected concentration of synthetic dopa-melanin used as a melanin standard [36,39].

### 2.4. Light Treatment and Cytotoxicity Assay

The day after the last feeding, ARPE-19 cultures were washed with Hanks’ balanced salt solution containing calcium and magnesium ions (HBSS) and irradiated for 1, 2, or 3 h at room temperature employing a dedicated Fully Reflective Solar Simulator-SSUV1.6KW(Sciencetech Inc, London, ON, Canada) equipped with a band-pass filter (350–780 nm) and blue dichroic color filter (360–502 nm) (Thin Film Imaging Technologies, Inc, Greenfield, MA, USA). The fluence rate, at the sample position, in the spectral region 360–502 nm was 8.3 mW/cm^2^.

Cytotoxicity mediated by MLF granules in ARPE-19 cells irradiated with 360–502 nm light was analyzed by MTT assay as described previously [40]. In brief, all culture wells were incubated with MTT solution in MEM with 10% FBS (final concentration 0.5 mg/mL) for 0.5 h at 37 °C. After incubation the resultant blue formazan precipitate was solubilized in DMSO/ethanol (1:1) mixture followed by reading the absorbance at 560 nm in a plate reader (GENios Plus, Tecan, Austria GMBH). Results were reported as a percentage of paired untreated controls. The cytotoxicity experiments were repeated a minimum of four times.

### 2.5. POS Isolation and Phagocytosis Assay

For analysis of phagocytic activity of ARPE-19 cells, fluorescently labelled cow photoreceptor outer segments (POS) were delivered to cell cultures. POS were isolated from fresh cow retinas according to modified techniques of Papermaster and Kennedy [41,42]. Briefly, extracted fresh bovine retinas were homogenized in PBS with 0.73 M sucrose on magnetic stirrer at 4 °C, under dim red light, and large pieces of retina were removed after few centrifugation (a. 173 g/6 min; b.173 g/6 min; c. 561 g/7 min; d. 5242 g/10 min). Collected pellets were layered on the top of three-step sucrose gradient (0.84; 1.0; 1.14 M) and centrifuged at 141,000× *g* for 1 h at 4 °C. POS fraction identified as reddish band at the 0.84/1.0 M interface was collected, washed in tris buffer, counted, and fluorescently labelled with FITC by incubation of 10^9^ POS/mL in PBS containing 10 μg/mL FITC for 1 h at room temperature in the dark by the method of McLaren [43] as described previously [44]. Delivery of 3.2 × 10^8^ POS-FITC/mL to irradiated controls and MLF containing cultures, immediately after irradiation or 24 h after induction of photic stress, was done according to a previously described method, with POS-FITC internalization proceeding over 5.5 h [13,45]. After this time, cultures were detached by trypsinization and suspended in PBS with 10% FBS. The percentage of fluorescence-positive events (Ex/Em: 488/525 nm) from 10,000 unfixed cells in suspension of cells was analyzed on a FACSCalibur instrument (BD Biosciences, San Jose, CA, USA) using CellQuest (BD Biosciences) software. Data were expressed as normalized phagocytosis (±SD) for paired control (untreated) and blue light irradiated cultures. The rate of phagocytosis by untreated cells was taken as 100%. Phagocytosis assays were repeated three times.

### 2.6. Coumarine Boronic Assay for Protein Peroxidation Products

Photooxidation of proteins in ARPE-19 cultures subjected to photic stress mediated by MLF granules isolated from donors of different age and MLF granules supplemented with antioxidants was monitored by coumarin boronic acid (CBA) assay according to modified procedure described previously [36,40,46]. In brief, lysates were prepared form control and blue light treated cultures containing MLF granules with or without antioxidants by passing cell pellets suspended in buffer containing DTPA (0.1 mM) and catalase (100 U/mL) through a needle. The lysates were transferred into 96-well black plates, and mixed with phosphate buffer (50 mM, pH 7.4) containing catalase (100 units/mL), DTPA (0.1 m*M*), and CBA (0.8 m*M*). The fluorescence intensity of the formed 4-hyroxycoumarin (COH), product of the interaction of CBA with protein hydroperoxides, was measured at 10 min intervals by plate reader (ClarioStar, BMG-Labtech, Knoxville, TN, USA), (Ex/Em: 360 nm/465 nm) for 20 h. The experiments were repeated three times.

### 2.7. Photoinduced Oxygen Consumption 

Time-dependent changes in oxygen concentration of control and irradiated samples were measured by electron paramagnetic resonance (EPR) oximetry, using a dissolved oxygen-sensitive spin probe mHCTPO (4-protio-3-carbamoyl- 2,2,5,5 tetraperdeuteromethyl-3-pyrrolin-1-yloxy) at 0.1 mM concentration as described elsewhere [14,38]. Briefly, samples containing melanolipofuscin granules (1 × 10^9^ granules/mL) or MLF supplemented with antioxidants in PBS in a standard EPR flat quartz cell was placed in resonant cavity equipped with an optical window and oxygen consumption was analyzed during irradiation of samples with violet light (400 nm; 20 mW/cm^2^) derived from a 50 W diode array illuminator (High Power UV Purple LED Chip, Chanzon, China). The EPR measurements were conducted by employing a Bruker EMX–AA EPR spectrometer (Bruker, BioSpin, Rheinstetten, Germany) operating at 9.5 GHz with 100-kHz field modulation. EPR spectra were registered using the following instrument settings: 1.06 mW microwave power, 0.006 mT modulation amplitude, 337.67 mT center field, 0.3 mT scan width, 40.960 ms time constant, and 5.24 s scan time. All oxygen uptake experiments were repeated at least two times.

### 2.8. Detection of Superoxide Anion by EPR-Spin Trapping

Light-induced formation of superoxide anion was monitored by EPR-spin trapping using 100 mM DMPO (5,5-Dimethyl-1-Pyrroline N-oxide) [47,48]. The suspension of melanolipofuscin granules (1 × 10^8^ granules/mL) or MLF supplemented with combination of antioxidants in 75% DMSO was irradiated employing the same light source as that described above for oxygen photoconsumption. The EPR samples were run using the following parameters: 10.6 mW microwave power, 0.05 mT modulation amplitude, 339 mT center field, 8 mT scan field, and 84 s scan time. The EPR-spin trapping measurements were performed employing the spectrometer described above. Simulations of EPR spectra were performed using WinSim2002 Software (National Institute of Environmental Health Sciences, Research Triangle Park, NC, USA). EPR spin trapping measurements were carried out three times yielding very similar results.

### 2.9. Direct Detection of Singlet Oxygen by Time-Resolved Near-Infrared Luminescence

Melanolipofuscin granules (2 × 10^8^ granules/mL or 5 × 10^7^ granules/mL) suspended in D_2_O phosphate-buffered (pD 7.4, 10 mM) with 1% DDM in a 1-cm-optical path quartz fluorescence cuvette (QA-1000; Hellma, Mullheim, Germany) were excited by light pulses generated by an integrated nanosecond DSS Nd:YAG laser system equipped with a narrow band-width optical parametric oscillator (NT242-1k-SH/SFG; Ekspla, Vilnius, Lithuania) [13,36]. The laser system delivered pulses at 1 kHz repetition rate, with pulse energy up to several hundred microjoules in the visible region, and up to several tens of microjoules in the UVA–UVB region. The ability of MLF granules, isolated from younger and older human donors, to photogenerate singlet oxygen was examined in the spectral range 320–550 nm. Some of the experiments were conducted using MLF granules preincubated with combination of 10µM zeaxanthin (ZEA) and 100 µM α-tocopherol (TOC). Data were collected using a computer-mounted PCI-board multichannel scaler (NanoHarp 250; PicoQuant GmbH, Berlin, Germany). Data analysis, including first-order luminescence decay fitted by the Levenberg–Marquardt algorithm, was performed by custom-written software. To minimize the effect of nonspecific fluorescence and laser light scatter, only longer-lived luminescence was analyzed (typically between 25 and 250 µs). Acquisition time for obtaining singlet oxygen phosphorescence signals was 20–60 s, depending upon experiment and MLF granules concentrations.

### 2.10. Immunofluorescence Analysis of the Cytoskeleton Organization of Cells Subjected to Photic Stress

For the analysis of cell cytoskeleton, ARPE-19 cells were seeded on microscope slides in 48-well plates at a density of 10^5^ cells/cm^2^. Cells were fed MLF_18-29 or MLF_50-59 granules with or without supplementation with antioxidants and immediately after induction of photic stress the cells were fixed with 3.7% formaldehyde, permeabilized with 0.1% Triton X-100, and blocked with 1% BSA. Subsequently, cells were immunostained with mouse monoclonal anti-human-tubulin IgG (Sigma-Aldrich) and Alexa Fluor 488-conjugated goat anti-mouse IgG (Life Technologies), and counterstained with Alexa Fluor 568-phalloidin (Life Technologies) and Hoechst 33342 (Sigma-Aldrich). Images of the cell cytoskeleton were obtained with scanning laser confocal microscope (LSM 900 with Airyscan 2, Carl Zeiss AG, Jena, Germany). Images were analyzed using ZEN Blue software (Carl Zeiss AG). Detailed description of the analysis can be found elsewhere [49].

## 3. Results and Discussion

### 3.1. Effect of Antioxidant Supplementation on Aerobic Photoreactivity of Melanolipofuscin Granules from Younger and Older Donors

After ultracentrifugation in discontinuous sucrose density gradient, melanolipofuscin granules, localized at the 1.6 M and 1.8 M interface, above the melanosome fraction (illustrated in Figure 1), consistent with the postulated complex structure of the granules that contain both melanin and lipofuscin components [19]. Indeed, a high resolution autofluorescence imaging of human RPE granules using structured illumination microscopy, revealed that melanolipofuscin granules contained a hyperfluorescent ring (presumably of lipofuscin origin) surrounding a darker melanin-like center core [50]. Additionally, a recent study, in which near-infrared autofluorescence and electron microscopy were used to detect and characterize pigment granules in RPE cell, suggested that melanolipofuscin originated from lipofuscin fusion with oxidatively degraded melanin [51]. Although RPE cells were pooled from the same number of younger and older donors, about 17–20% more MLF granules was obtained in the group of older donors, compared to younger donors. Differences in pigmentation of MLF granules between both age groups were observed visually, and confirmed by EPR spectroscopy as discussed below. Here, we analyzed photoreactivity of MLF granules isolated from RPE of younger (MLF_18-29) and older (MLF_50-59) human donors, and examined the effects of zeaxanthin and vitamin E. First, the ability of the pigment granules to induce oxygen photoconsumption was compared. Although oxygen consumption accompanying a photoreaction provides limited information about the photoreaction mechanism, it is a convenient indicator of oxygen-dependent reactivity [52].

As shown in Figure 2a,b, irradiation of melanolipofuscin suspension with blue light was accompanied by oxygen uptake, which depended on MLF donors age and supplementation with antioxidants. Comparison of MLF from younger and older donors revealed that the observed rate of photoinduced oxygen uptake was three-fold faster in suspension of MLF from older donors than that of MLF isolated from younger donors. Supplementation of MLF granules with combination of zeaxanthin and α-tocopherol reduced the observed rate of oxygen photoconsumption by a factor of 4.7 in case of MLF_18-29, and by factor of 2.2 in case of MLF_50-59 (Figure 2a,b).

Considering that blue light-irradiation of both lipofuscin granules and melanosomes induced, with different efficiency, the formation of superoxide anion [13,14,18,36,53], we examined the ability MLF isolated from younger and older donors to photogenerate this semireduced oxygen species employing EPR-spin trapping. Data shown in Figure 3a–c reveal accumulation of the DMPO-OOH spin adduct during blue light irradiation of MLF. Although the signal to noise of the detected spin adduct is relatively weak, simulation of its EPR spectrum, assuming the following spectral parameters: a_N_ = 1.2995 mT, a_H1_ = 1.0458 mT, and a_H2_ = 0.1335 mT, confirmed that superoxide anion formed the detected spin adduct with DMPO. The calculated rates of DMPO-OOH accumulation indicate that MLF_50-59 photogenerated superoxide anion 1.2 times faster than MLF_18-29. The data also show that zeaxanthin and α-tocopherol supplementation only partially inhibited the photogeneration of superoxide anion. Thus, preincubation of MLF with antioxidants lowered the observed rates of the DMPO-OOH accumulation by factor of 1.4 for MLF_18-29 + ZEA/TOC and by factor of 1.2 for MLF_50-59 + ZEA/TOC, when compared to the nonsupplemented granules. Although the rate of oxygen photoconsumption previously determined for purified lipofuscin granules was about six-fold higher than that of melanosomes, an inverse relation was observed for the corresponding accumulation of hydrogen peroxide [18]. It was concluded that in lipofuscin mediated photoprocesses, unlike in photoreactions involving melanosomes, only a small fraction of the photoconsumed dioxygen was converted into hydrogen peroxide, suggesting that these pigment granules could undergo different types of photochemistry. The complex structure of MLF may justify mix photochemistry of the granules, in which the melanin component is mostly involved in free radical processes leading to generation of superoxide anion and hydrogen peroxide [54,55,56], while the lipofuscin component via energy transfer generates singlet oxygen [13,14,53]. Importantly, singlet oxygen being significantly more reactive than superoxide anion and hydrogen peroxide could lead to secondary reactions such as oxidation of unsaturated lipids and proteins [52].

The ability of MLF to photogenerate singlet oxygen was examined by direct detection of singlet oxygen, based on time-resolved measurements of near-infrared phosphorescence [52]. Even though this method is very sensitive and specific, its application to highly dispersive and intrinsically unstable samples such as aqueous suspension of MLF granules is quite challenging. To homogenize and stabilize melanolipofuscin suspension in D_2_O, the measurements were carried out after addition of 1% n-dodecyl-β-D-maltopyranoside (DDM). This mild detergent was successfully used in our previous studies to determine photogeneration of singlet oxygen by lipofuscin and melanin granules [13,36]. Representative data of the detected 1270 nm phosphorescence, shown in Figure 4a, indicate that MLF from younger and older donors photogenerate almost equal fluxes of singlet oxygen when excited 420 nm. The singlet oxygen nature of the observed phosphorescence was confirmed by control experiments, in which the long-lived phosphorescence signal was practically eliminated by saturating the samples with argon (Figure 4b) and significantly quenched by addition of 0.5 mM azide, an efficient physical quencher of singlet oxygen (Figure 4c). Substantial quenching of the singlet oxygen phosphorescence was also observed after enrichment of MLF granules with 10 µM ZEA and 100 µM TOC (Figure 4d,e). This is to be expected in view of the reported very high efficiency of carotenoids to quench singlet oxygen [32]. Thus, the data suggest that the inherent photoreactivity of MLF could be substantially reduced by natural antioxidants such as zeaxanthin and vitamin E. 

The efficiency of MLF granules to photogenerate singlet oxygen was analyzed at selected wavelength between 590 and 320 nm. Figure 4f shows action spectra, i.e., wavelength dependence of singlet oxygen intensity normalized to equal laser energy. A noticeable increase in the photogeneration of singlet oxygen by MLF is observed in the violet part of the spectrum (between 450 and 400 nm), while below 400 nm the efficiency of singlet oxygen photogeneration increases even more rapidly. This wavelength dependence coincides with the determined action spectrum of singlet oxygen photogeneration by lipofuscin granules [13], confirming the lipofuscin contribution to singlet oxygen dependent photoreactivity of melanolipofuscin granules.

The obtained results demonstrate significant aerobic photoreactivity of melanolipofuscin granules, which upon photoexcitation with short-wavelength visible light and near UV generate reactive oxygen species that in vivo could be involved in chronic oxidative stress of the outer retina.

### 3.2. Effect of Antioxidant Supplementation on Photoperoxidation of Cellular Proteins Mediated by Melanolipofuscin Granules from Younger and Older Donors

A possible consequence of aerobic photoreactivity of melanolipofuscin could be oxidation of key cellular targets such as proteins, nucleic acid, and membrane lipids. Here, we examined whether measurable photooxidation of cellular proteins could be detected in ARPE-19 cultures fed MLF granules isolated from human RPEs of two selected age groups. Figure 5 shows that blue light irradiation of ARPE-19 cultures fed MLF_18-29 or MLF_50-59 caused significant increase in 4- hydroxycoumarin (COH) fluorescence (product of the interaction of coumarin boronic acid with protein hydroperoxides), with the more prominent effect being observed for MLF from older donors. Although the fluorescence signal was also observed in lysates of control light-treated cultures without MLF, and in nonirradiated controls fed MLF_18-29 or MLF_50-59 (Figure 5a,c), the signal intensity of the control samples was significantly lower than that in irradiated samples containing MLF. The initial relative rates of COH fluorescence were as follows: 84.6 ± 22.4 (control dark), 74.2 ± 19.5 (MLF 18-29 dark), 63.1 ± 15.1 (MLF 50-59 dark), 471 ± 24 (control light), 676.2 ± 48.4 (MLF 18-29 light), 2347.7 ± 82.8 (MLF 50-59 light) (Figure 5d). 

The results demonstrate that melanolipofuscin granules after phagocytosis by ARPE-19 cells were able to photoinduce oxidation of cellular proteins. The pro-oxidizing effect of MLF increases with age of human donors. It is worth noticing that supplementation of MLF granules with zeaxanthin and α-tocopherol reduced almost in half the extent of protein oxidation mediated by MLF from both age groups (Figure 5d). The inhibitory effect of supplementation of MLF with antioxidants on photooxidation of cellular proteins and the efficient quenching of singlet oxygen (vide supra) by the antioxidants, suggest Type II photochemistry of melanolipofuscin, with a major involvement of singlet oxygen. The apparent disparity between the efficiency of quenching of singlet oxygen by antioxidants and their inhibitory effect on photooxidation of cellular proteins, could result from different content of the antioxidants in MLF granules expected in model systems and in cells after phagocytosis of the granules. 

### 3.3. Content of Melanolipofuscin Granules in ARPE-19 Cells after Feeding the Cells MLF Granules Isolated from Younger and Older Donors

Our previous flow cytometry studies demonstrated that phagocytic uptake of LF_18-29 and LF_50-59 by ARPE-19 cells was at the same level [13]. Similar conclusion was reached comparing phagocytic uptake of melanosomes isolated from younger and older donors [21]. It is important to stress that melanolipofuscin granules, due to their melanin component, exhibit EPR signals with parameters similar to those of RPE melanosomes [57]. Therefore, this noninvasive technique was used here to compare the ability of ARPE-19 cultures to phagocytose MLF granules isolated from younger and older donors. Figure 6 shows EPR spectra of MLF granules (10^9^/mL) isolated from younger and older donors and of ARPE-19 cells fed MLF_18-29 and MLF_50-59. The figure also shows EPR signal of synthetic dopa-melanin used as eumelanin standard [36]. Although the integrated intensities of these EPR signals differ substantially, comparison of signals obtained for cells containing each type of MLF granules and signals obtained for isolated MLF_18-29 and MLF_50-59 granules, enabled us to determine that the number of internalized granules per cell was 15 ± 2.5, regardless of the donor’s age (Figure 6f). The results indicate that any age-related changes in the composition of MLF had no effect on phagocytic uptake of the granules. 

The detected difference in EPR signal intensities between similar number of MLF_18-29 granules and MLF_50-59 granules is rather surprising and deserves a comment. Assuming that the melanin component of melanolipofuscin from younger and older donors had similar paramagnetic properties, the concentration of melanin in MLF_18-29 can be calculated as 1.82 × 10^−10^ mg/per granule whereas in MLF_50-59 it was 3.01 × 10^−10^ mg/per granule. The higher content of melanin in melanolipofuscin from older donors may suggest that more melanin accumulates in melanolipofuscin granules with age. Perhaps the age-related photooxidation of melanosome, postulated by us [58,59], facilitates fusion of the photoaged melanosomes with lipofuscin.

### 3.4. Effect of Antioxidant Supplementation on Phototoxicity of ARPE-19 Cells Mediated by Melanolipofuscin Granules from Younger and Older Donors

Although phototoxicity of MLF in ARPE-19 cells has been reported by other researchers [19], this study focused on correlation between aerobic photoreactivity of MLF from donors of different age and sublethal phototoxicity of the granules relevant for chronic effects expected in vivo. We also aimed at examining if selected natural antioxidants could modify MLF-mediated phototoxicity. Several preliminary experiments were conducted using MTT assay to evaluate the dark cytotoxicity after feeding ARPE-19 cells MLF granules isolated from human donors of different age (Figure 7). Control analyses showed no dark cytotoxicity of either MLF granules (Figure 7a). As expected, no cytotoxicity was observed when control cells without pigment granules were irradiated with blue light for different time intervals (Figure 7a). On the other hand, a dose-dependent phototoxicity was apparent after blue light irradiation of cells containing each type of MLF granules, with the effect being stronger for MLF from older donors (Figure 7a). Irradiation of cells fed MLF_18-29 or MLF_50-59 for 2 h reduced their viability to 80% or 70% of controls, respectively (Figure 7a), and irradiation of the cells for 3 h reduced their survival to 59% in cultures containing MLF_18-29 and to 46% for cells fed MLF_50-59 (Figure 7a). Based on these results, lower light doses were selected for analysis of MLF-mediated effects on phagocytic activity of the cells and their cytoskeleton organization.

We have previously demonstrated that combination of zeaxanthin and α-tocopherol provided efficient protection to ARPE-19 cells against photodynamic stress mediated by photosensitizing dyes [35] or lipofuscin [13]. Here we analyzed if supplementation with combination of these two antioxidants could modify phototoxic potential of MLF granules. As shown in Figure 7b, preincubation of melanolipofuscin granules isolated from RPE of younger or older donors with a combination of 10 µM zeaxanthin and 100 µM α-tocopherol, prior to their uptake by the cells and light-treatment, provided significant cytoprotection. Thus, the survival of cells containing MLF_18-29, or MLF_50-59, supplemented with antioxidants was reduced only by 22% and 30%, respectively (Figure 7b). 

### 3.5. Effect of Antioxidant Supplementation on Specific Phagocytic Activity of ARPE-19 Cells Subjected to Photic Stress Mediated by Melanolipofuscin Granules from Younger and Older Donors

A major goal of this study was to examine if sublethal photic stress mediated in ARPE-19 cells by internalized melanolipofuscin granules could inhibit the cell phagocytosis of fluorescently labeled photoreceptor outer segments (POS). Figure 8a shows that blue light irradiation of cultures fed melanolipofuscin granules induced marked inhibition of the specific phagocytosis of POS membranes, with the effect being more significant in cultures containing MLF granules isolated from older donors (Figure 8a). Phagocytic activity of ARPE-19 cultures fed MLF decreased with the photic stress in a dose-dependent manner, reaching about 44% of nontreated controls in cells containing MLF_18-29 irradiated for 2 h (Figure 8a), and in cultures fed MLF_50-59, the observed POS uptake was reduced to about 32% of control (Figure 8a). It must be emphasized that the observed inhibition of the phagocytosis was transient; when POS were delivered to cells 24 h after induction of photic stress mediated by MLF granules, the efficiency of the cells to phagocytose POS substantially recovered (up to 78.5% for MLF_18-29 and up to 77.7% for MLF_50-59) (Figure 8b). Control experiments confirmed the photochemical nature of the MLF-mediated inhibition of the specific phagocytosis of ARPE-19 cells. Thus, neither MLF granules without light nor blue light treatment without MLF affected phagocytic activity of the cells (data not shown). 

The molecular mechanism of the observed inhibition of the specific phagocytosis, induced by photic stress mediated by melanolipofuscin, remains unclear. At least three possible mechanisms could be considered for explaining the photic damage of this sophisticated molecular machinery involved in POS phagocytosis. (1) The impairment in phagocytosis could be associated with melanolipofuscin-related oxidative modifications in receptor proteins involved in particle binding (integrin αvβ5) and engulfment (MerTK) [60,61,62,63]. This assumption is supported by results of our previous studies that have demonstrated substantial sensitivity of the receptor proteins to photic stress mediated by photosensitizing dyes such as merocyanine 540 or rose Bengal and the coincident transient reduction of integrin αvβ5 and MerTK receptor and inhibition of phagocytosis [64]. (2) Proper organization and function of cell cytoskeleton are required for efficient phagocytosis of POS membranes by RPE cells [65]; therefore, oxidative modifications of the cytoskeletal proteins such as β-actin is a plausible explanation of the detected inhibition of phagocytosis induced by melanolipofuscin mediated photic stress. Indeed, it was recently shown that inhibition of specific phagocytic activity of ARPE-19 cells induced by photic stress mediated by lipofuscin granules [13] coincided with significant disruption of the cell cytoskeleton organization and modification of the cell nanomechanical properties [15]. The observed photooxidation of cellular proteins, mediated by internalized melanolipofuscin (vide supra), is consistent with the postulated relationship between the reduction in phagocytic activity of POS and oxidative modification of cell cytoskeletal proteins. The disruption of the cell cytoskeleton, induced by melanolipofuscin-mediated photoreactions (vide infra), is in support of such a mechanism.

(3) As shown in an independent study, melanolipofuscin photoreactivity could lead to oxidation of lipids [18]. Therefore, the observed inhibition of POS phagocytosis could also result from MLF-mediated photooxidation of membrane lipids. This could affect the bilayer structure and modify various functional properties of the membrane [66,67,68,69,70]. There seems to be a distinct relationship between phagocytic activity of RPE cells and their lipid metabolism [71]. A correlation between the formation of cholesterol hydroperoxides (5α-OOH) and the degree of phagocytosis inhibition was observed in photodynamically treated RPE cells [35].

Numerous studies have supported the hypothesis that antioxidants protect the retina against damage resulting from oxidative stress [72,73,74,75]. However, it must be stressed that the impact of supplementation with specific antioxidants on progression of age-related macular degeneration remains unclear [76]. Here we sought to determine whether a combination of two important retinal antioxidants—zeaxanthin and α-tocopherol—confer protection of phagocytic function of ARPE-19 cells against photic stress mediated by MLF granules. Figure 9 shows that enrichment of melanolipofuscin granules with the antioxidants resulted in a much weaker inhibition of the specific phagocytosis, induced by photic stress, compared to cells containing internalized MLF without antioxidants. Flow cytometry analyses documented that the phagocytic activity of cells containing supplemented MLF granules isolated from younger donors (MLF_18-29) was only inhibited by 19% (Figure 9a) and by 20% by supplemented MLF granules isolated from older donors (MLF_50-59) (Figure 9b). The observed protection by zeaxanthin and α-tocopherol may be attributed to quenching of singlet oxygen and scavenging of oxidizing radicals that are generated as a result of photoactivation of melanolipofuscin granules. Therefore, it can be concluded that at least in vitro the two natural antioxidants exhibited significant protection against photic stress mediated by melanolipofuscin granules, which become the dominant pigment of the aging retinal pigment epithelium. 

Limitations of the commercially available ARPE-19 cell line as a model for AMD study are commonly discussed. ARPE-19 cells, like other stabilized cell lines, lose certain properties such as pigmentation, when compared to RPE in vivo. Furthermore, problems with limited epithelial maturation and complete polarization of these cells were also acknowledged. Despite these drawbacks, ARPE-19 cultures form morphologically and functionally polarized epithelial monolayers similar to those of RPE in vivo. Moreover, it was shown that this stable cell line is a convenient in vitro model for studying how oxidative stress affects various biological functions of these cells, including their ability to phagocytize POS discs. We have also found that appropriate cell culture conditions stimulate in ARPE-19 cells expression of αvβ5 integrin and MerTK receptor proteins, essential for efficient phagocytosis of POS discs [64]. Alternative models for in vitro study might be primary cultures from rodents or humans. However, these in vitro models are not without limitations either. It is worth emphasizing that widely used lines from rodents (e.g., the rat RPE-J line) exhibit species differences. On the other hand, primary cultures propagated from the RPE of adult human donor eyes are highly donor-to-donor variable making outcomes from such cultures less generalizable [77,78]. 

### 3.6. Effect of Antioxidant Supplementation on Cytoskeleton Organization of ARPE-19 Cells Subjected to Photic Stress Mediated by Melanolipofuscin Granules Isolated from Donors of Different Age

Proper organization of cytoskeleton of RPE cells is important for maintaining the characteristic hexagonal shape of the cells [79] and their phagocytic functions [3,4]. Therefore, we examined the effect of melanolipofuscin-mediated photic stress on organization of the cell cytoskeleton. Representative fluorescence images of cytoskeleton organization in ARPE-19 cells (actin and microtubules) are shown in Figure 10. In control cells, actin was mostly incorporated into thick fibers, whereas microtubules were abundant all over the cells. On the other hand, in cells containing MLF granules irradiated with light, both actin cytoskeleton and microtubules were significantly disrupted. In particular, actin formed thin fibers, which were less abundant than in control cells, whereas microtubules were completely disrupted, suggesting their higher vulnerability to photic stress mediated by MLF than that of the actin cytoskeleton. The effect was stronger in cells containing granules of the older age group. Importantly, supplementation of the granules with antioxidants resulted in a significant protection of the cell cytoskeleton, i.e., organization of both actin and microtubules of the irradiated cells resembled that of control cells. The results indicate that excitation of MLF granules with UV-blue light leads to substantial modifications of the cell cytoskeleton, which is the main contributor to the mechanical properties of the cells [80]. We have recently demonstrated that RPE plays a fundamental role in mechanoprotection of the blood–retina barrier [7]. Disruption of this barrier during ageing leads to different pathological conditions [81]. Results obtained in this study indicate protective properties of antioxidants in sustaining the mechanical properties of ARPE-19 cells under photic stress mediated by MLF granules.

## 4. Conclusions

Melanolipofuscin, one of the major age pigments of human retinal pigment epithelium, exhibits significant aerobic photoreactivity that could be explained by photochemical properties of lipofuscin and melanin components of the age pigment. Singlet oxygen and superoxide anion photogenerated by melanolipofuscin induce oxidation of cellular proteins and may be involved in phototoxic reactions that disturb the organization of cytoskeleton of RPE cells and inhibit their phagocytic activity. While the phototoxic effects of melanolipofuscin increase with aging, zeaxanthin and α−tocopherol could at least partially the reduce the photoreactivity and phototoxicity of the age pigment.

## Figures and Tables

**Figure 1 antioxidants-09-01044-f001:**
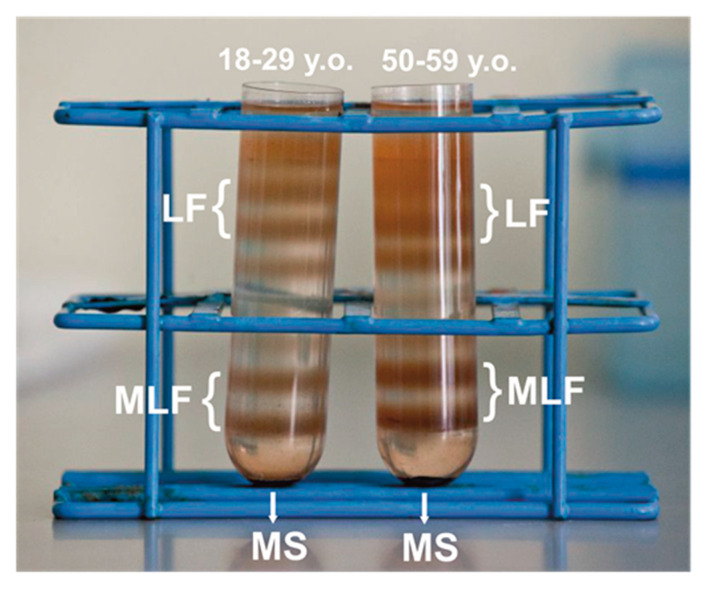
Melanolipofuscin (MLF) granules identified as two orange-brown fractions in discontinuous sucrose density gradient after centrifugation of retinal pigment epithelium (RPE) homogenates isolated from younger (18–29 y. o.) or older (50–59 y. o.) human donors. MLF granules containing less melanin formed fraction at the 1.6 M/1.8 M interface, while MLF granules containing more melanin formed a heavier fraction at the 1.8 M/2.0 M interface. The heaviest melanosomes fraction (MS) and the lightest lipofuscin fractions (LF) are also shown in the picture.

**Figure 2 antioxidants-09-01044-f002:**
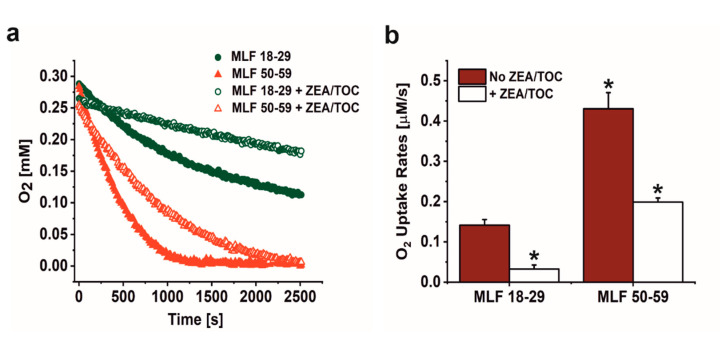
The effect of antioxidants on oxygen uptake induced by irradiation of melanolipofuscin (MLF) granules isolated from younger (18–29 y. o.) and older (50–59 y. o.) human donors monitored by EPR oximetry. (**a**) Oxygen uptake in irradiated MLF samples isolated from donors: 18–29 y. o. (solid, green circle) or 50–59 y. o. (solid, red triangle) and in suspension of granules from younger donors (open, green circle) and from older donors (open, red triangle), enriched with combination of antioxidants. (**b**) Calculated rates of light induced oxygen uptake in suspensions of MLF with no antioxidants (no ZEA/TOC) (brown bar) and samples supplemented with mixture of zeaxanthin and α-tocopherol (+ ZEA/TOC) (white bar). Values are means of three replicate samples (error bars indicate SD (*n* = 3)). Outcomes differed significantly between irradiated MLF samples from each donors group (18–29 y. o. and 50–59 y. o.) and between light treated MLF_18-29 and MLF_50-59 without and with antioxidants, as indicated by asterisks (*t-*test analyses, *p* < 0.05).

**Figure 3 antioxidants-09-01044-f003:**
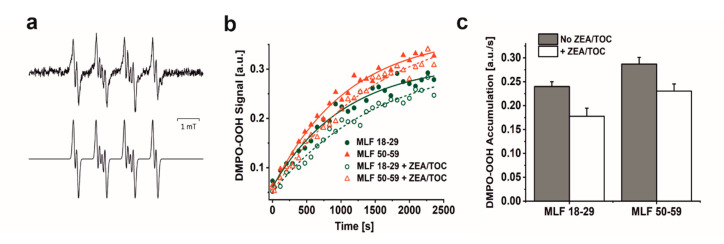
The effect of antioxidants on photogeneration of superoxide anion in suspension of melanolipofuscin (MLF) granules isolated from younger (18–29 y. o.) and older (50–59 y. o.) detected by EPR spin trapping using DMPO as a spin trap. (**a**) Representative EPR spectra of DMPO-OOH spin adducts recorded in irradiated MLF_50-59 samples (upper signal) and simulated EPR spectrum for MLF_50-59 sample (lower signal); simulation parameters: a_N_ = 1.2995 mT, a_H1_ = 1.0458 mT, a_H2_ = 0.1335 mT, and R^2^ = 0.935. (**b**) Kinetics of the DMPO-OOH spin adduct formation during blue light illumination of MLF samples isolated from donors: 18–29 y. o. (solid, green circle) or 50–59 y. o. (solid, red triangle) and in suspension of granules from younger donors (open, green circle) and from older donors (open, red triangle), enriched with combination of antioxidants. (**c**) Calculated rates of light induced accumulation of DMPO-OOH spin adduct in suspensions of MLF with no antioxidants (no ZEA/TOC) (brown bar) and samples supplemented with mixture of zeaxanthin and α-tocopherol (+ ZEA/TOC) (white bar). Values are means of three replicate samples (error bars indicate SD (*n* = 3)). Outcomes differed significantly between irradiated MLF samples from each donors group (18–29 y. o. and 50–59 y. o.) and between light treated MLF_18-29 and MLF_50-59 without and with antioxidants, as indicated by asterisks (*t-*test analyses, *p* < 0.05).

**Figure 4 antioxidants-09-01044-f004:**
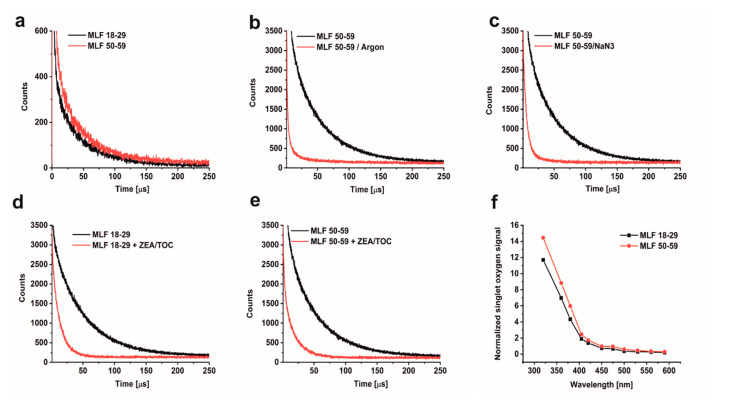
Time-resolved 1270 nm luminescence induced by photoexcitation of melanolipofuscin (MLF) granules suspended in D_2_O, after 24 h incubation with 1% DDM. (**a**) Luminescence decay detected in MLF_18-29 y. o. samples (lower trace) and or MLF_50-59 y. o. samples (upper trace) (granule concentration 5 × 10^7^/mL). (**b**) Luminescence decay detected in MLF_50-59 y. o. samples (upper trace) and in argon-saturated MLF_50-59 y. o. samples (lower trace), (granule concentration 2 × 10^8^/mL). (**c**) Luminescence decay detected in oxygen-saturated MLF_50-59 y. o. samples before (upper trace) and after addition of 0.5 mM NaN_3_ (lower trace), (granule concentration 2 × 10^8^/mL). (**d**) Luminescence decay detected in MLF_18-29 y. o. samples before (upper trace) and after addition antioxidants (10 µM ZEA and 100 µM TOC) (lower trace), (granule concentration 2 × 10^8^/mL). (**e**) Luminescence decay detected in MLF_50-59 y. o. samples before (upper trace) and after incubation with antioxidants (10 µM ZEA and 100 µM TOC) (lower trace), (granule concentration 2 × 10^8^/mL). (**f**) Action spectrum of singlet oxygen photogeneration for melanolipofuscin granules from younger donors (lower trace) or older donors (upper trace); initial intensities of singlet oxygen luminescence at 1270 nm were normalized to equal laser power at the indicated wavelength, (granule concentration 5 × 10^7^/mL). MLF granules were excited by 420 nm laser pulses. Graphs show data from representative samples.

**Figure 5 antioxidants-09-01044-f005:**
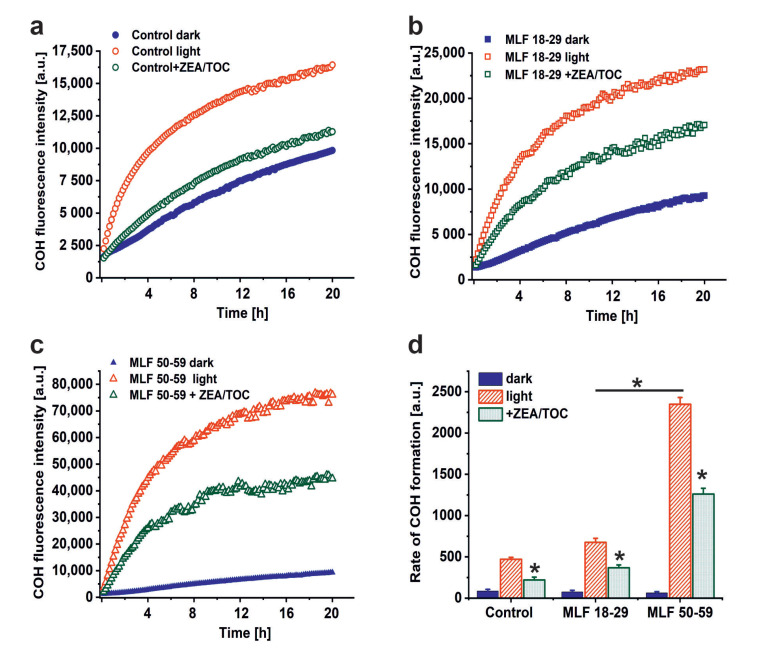
The effect of antioxidants on melanolipofuscin-mediated oxidation of proteins in ARPE-19 cells. (**a**) Time course of 4-hydroxycoumarin (COH) fluorescence measured in lysates of nonirradiated control ARPE-19 cells (solid, blue circle), blue light treated control cultures (open, red circle), or control cells supplemented with zeaxanthin and α-tocopherol (ZEA/TOC) (open, green circle). (**b**) Time course of COH fluorescence measured in lysates of nonirradiated ARPE-19 cultures fed with MLF_18-29 y. o. (solid, blue square), blue light treated ARPE-19 cells after feeding with MLF_18-29 y. o. (open, red square), or blue light treated cultures fed with MLF_18-29 y. o. supplemented with zeaxanthin and α-tocopherol (ZEA/TOC) (open, green square). (**c**) Time course of COH fluorescence measured in lysates of nonirradiated ARPE-19 cultures fed with MLF_50-59 y. o. (solid, blue triangle), blue light treated ARPE-19 cells after feeding with MLF_50-59 y. o. (open, red triangle), or blue light treated cultures fed with MLF_50-59 y. o. supplemented with zeaxanthin and α-tocopherol (ZEA/TOC) (open, green triangle). (**d**) Calculated rates of COH formation (initial increase of the fluorescent intensity normalized to time) for all tested samples. Values are expressed as means ± SD (*n* = 3). Slopes and the maximum levels of the detected COH fluorescence differed between cells containing MLF_18-29 and MLF_50-59 granules, and between light-treated cells fed both types of MLF versus nonirradiated cultures with MLF granules and between samples containing MLF granules without and with added combination of zeaxanthin and α-tocopherol (ZEA/TOC) (GraphPad Prism five slope analysis, *p* < 0.0001). The rates of COH fluorescence evolution differed significantly for all tested samples (one-way ANOVA followed by post hoc analysis with Tukey test and Bonferroni test (*p* < 0.05).

**Figure 6 antioxidants-09-01044-f006:**
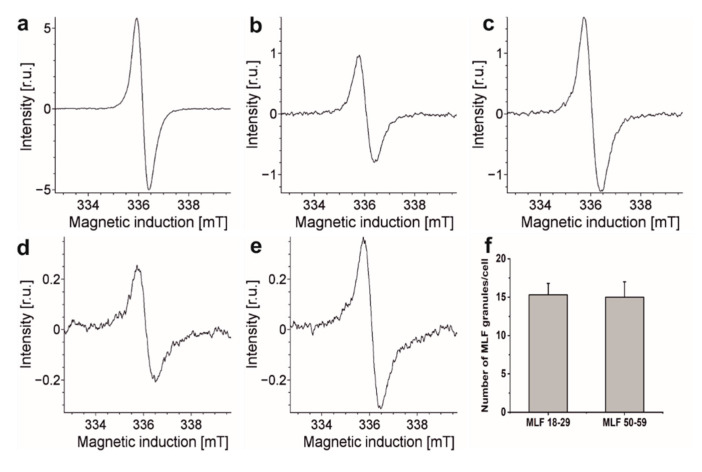
Representative EPR spectra recorded for: (**a**) synthetic dopamelanin (0.5 mg/mL), (**b**) melanin in melanolipofuscin granules isolated from younger human donors (18–29 years old) (10^9^/mL), (**c**) melanin in melanolipofuscin granules isolated from older human donors (50–59 years old) (10^9^/mL), (**d**) melanin in 7.34 × 10^6^ ARPE-19 cells fed MLF_18-29 granules, and (**e**) melanin in 5.77 × 10^6^ ARPE-19 cells fed MLF_50-59 granules. (**f**) Calculated number of MLF_18-29 or MLF_50-59 granules per cell. All samples were measured in liquid nitrogen using a standard finger-type quartz dewar. All measurements were repeated three times.

**Figure 7 antioxidants-09-01044-f007:**
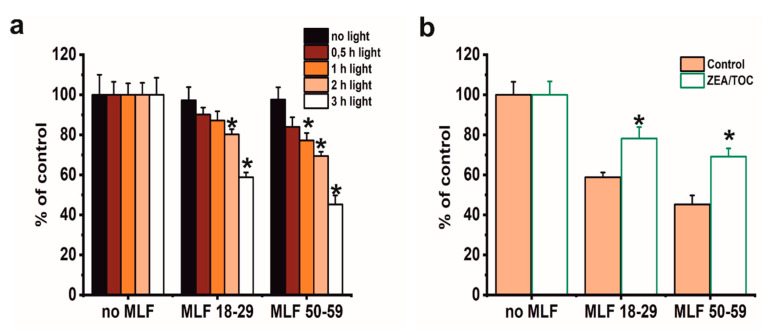
The effect of antioxidants on metabolic activity of light-treated ARPE-19 cells containing phagocytized MLF granules isolated from human donors of different age determined by MTT assay. (**a**) The effect of photic treatment of ARPE-19 cells fed MLF_18-29 or MLF_50-59. Control cultures or cells fed melanolipofuscin granules isolated from younger donors (age: 18–29 years old) or melanolipofuscin granules isolated from older donors (age: 50–59 years old) were either kept in the dark (black bar) or irradiated with blue light for 0.5 h (brown bar), 1 h (orange bar), 2 h (pink bar), or 3 h (white bar) and analyzed by the MTT assay 24 h after treatment. (**b**) The effect of supplementation with antioxidants on cytotoxicity in ARPE-19 cells subjected to photic stress mediated by phagocytized MLF granules. Control cultures without MLF or cells fed MLF granules with no antioxidants (pink bar) and cultures enriched with zeaxanthin and α-tocopherol or preloaded with MLF granules enriched with antioxidants (white bar) were irradiated with blue light for 3 h. Then, 24 h after light treatment MTT assay for cell metabolic activity was done. Data are presented as the percentage of control nonirradiated cells. Values are expressed as means ± SD (*n* = 4). Asterisks indicate significant differences between blue light treated ARPE-19 cultures containing human MLF granules from donors of different age and dark controls, and significant differences between nonsupplemented and antioxidant-enriched samples (*t-*test analyses, *p* < 0.05).

**Figure 8 antioxidants-09-01044-f008:**
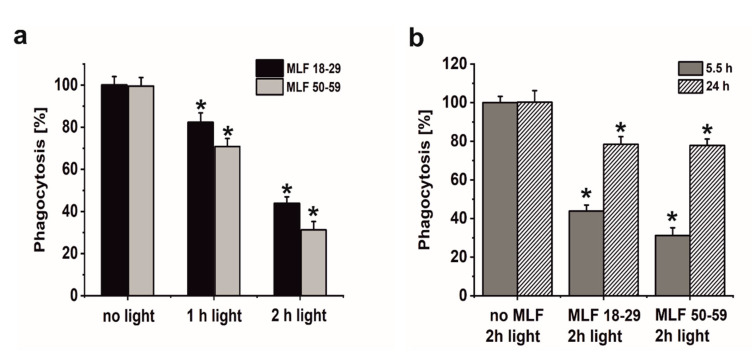
The effect of light stress mediated by melanolipofuscin granules on specific phagocytic activity of ARPE-19 cells determined by flow cytometry. (**a**) Dose-dependent inhibition of specific phagocytic activity of irradiated cells fed MLF_18-29 (black bar) or MLF_50-59 (grey bar). (**b**) Time-dependent recovery of the inhibited phagocytic activity of ARPE-19 cultures, induced by photic stress, mediated by phagocytized MLF_18-29 or MLF_50-59 granules analyzed by flow cytometry 5.5 h (grey bar) or 24 h (crossed bar) after the treatment. Control cultures or cells containing MLF granules were either kept in the dark or were exposed to blue light for selected time intervals. POS-FITC were delivered to cultures at density 3.2 × 10^8^ particles/mL and cytometric analyses were done 5.5 and 24 h after the treatment. Obtained results were normalized to control nontreated cells. Values are means ± SD (*n* = 3). POS-FITC uptake by light-treated cultures differed significantly comparing MLF granules from younger donors and older donors at both irradiation time points (**a**) and between time points 5.5 and 24 h after induction of oxidative stress mediated by MLF granules (**b**) as indicated by asterisks (*t*-test analyses, *p* < 0.05).

**Figure 9 antioxidants-09-01044-f009:**
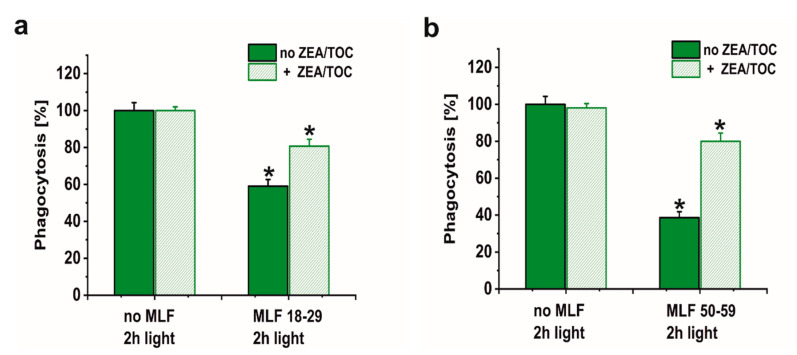
The effect of antioxidants on specific phagocytic activity of light-treated ARPE-19 cells containing phagocytized MLF granules isolated from human donors of different age determined by flow cytometry. (**a**) POS-FITC uptake by irradiated control cultures and cells containing MLF_18-29 granules without (green bar) or with combination of zeaxanthin and α-tocopherol (crossed green bar). (**b**) Phagocytosis of POS-FITC by light-treated control cultures or cells fed MLF_50-59 granules without (green bar) or with combination of zeaxanthin and α-tocopherol (crossed green bar). Control cells without MLF granules and cells fed MLF particles with no antioxidants or cultures enriched with zeaxanthin and α-tocopherol or preloaded with MLF granules with added antioxidants were irradiated with blue light. POS-FITC were delivered to ARPE-19 cultures at density 3.2 × 10^8^ particles/mL and phagocytosis was quantified by flow cytometry. Data were normalized to control cultures lacking granules. Values are means ± SD (*n* = 3). POS-FITC uptake differed significantly between control and antioxidant treated cultures as indicated by asterisks (*t*-test analyses, *p* < 0.05).

**Figure 10 antioxidants-09-01044-f010:**
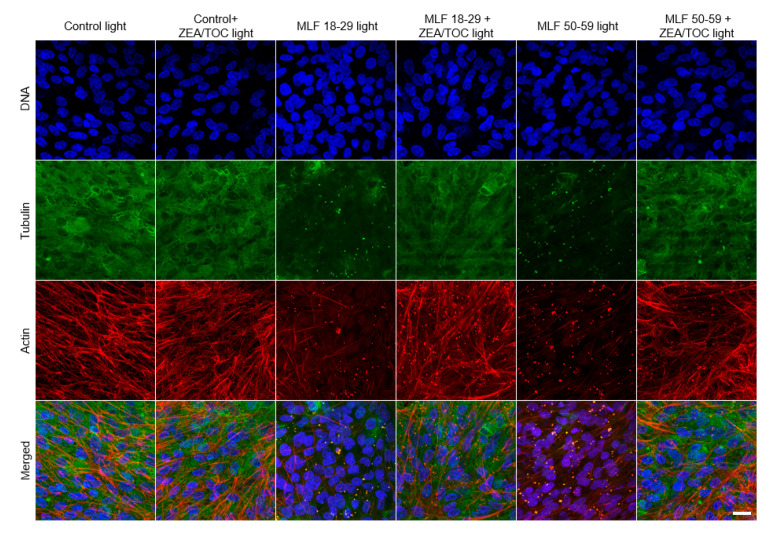
Effect of melanolipofuscin-mediated photic stress on the cytoskeleton of ARPE-19 cells. Fluorescence images of the cells’ cytoskeletons are shown in the maximum intensity projection mode. Individual melanolipofuscin granules are apparent in cells fed MLF. Scale bar for all images represents 20 µm.

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
