# Peer review of "The Effect of Antioxidants on Photoreactivity and Phototoxic Potential of RPE Melanolipofuscin Granules from Human Donors of Different Age"

_antioxidants, 2020, doi:10.3390/antiox9111044_

Round 1

Reviewer 1 Report

The manuscript by Olchawa et al demonstrates that the antioxidants zeaxanthin and alpha-tocopherol partially prevented the harmful effects caused by melanolipofuscin granules isolated from human eyes. Generally, the manuscript is clear and the data is well presented. However, several aspects need to be clarified before this manuscript can be accepted for publication.

Minor:

  1. The manuscript needs to be carefully edited to correct some grammar and spelling errors. Hereby some examples are presented:
  • Line 86: “enrichement of pigmet” should be “enrichment of pigments”
  • Line 140: “APRE-19” should be “ARPE-19”
  • Line 221: “supplenemtation” should be “supplementation”
  • Line 401: “signals” should be “signals”
  • Line 36: “is the outermost part of the eye retina”, the sentence should be rephrased
  • Line 41: “Recently,” refers to work published in 2017 and therefore not very recent
  •  

Major:

  1. The material and methods section needs to be re-written to allow others to better understand what was done and how was it done. This section has too many references what reduces the readability of the paper.
  • Line 66: Materials sections might be removed, and the materials used mentioned in the different following sections.
  • Line 86: Isolation of the granules: The information of how RPE was isolated, from how many eyes, the characteristics of the donors and the post-mortem time from the death until the isolation is lacking and needs to be added to the manuscript.
  • Line 94: The authors need to add details of how the pellets were obtained.
  • Line 99 and 104: It is quite difficult to understand how the granules were used; the authors should add a scheme explaining it.
  • Line 100: “injected” should be “added”.
  • Line 110: The information about the type of dishes used is not informative and it should be removed. However, it should be clear the culture medium used in each experiment.
  • Line 139: How the POS were isolated needs to be clearly stated.
  1. One of the main issues of this work is related to sampling. No information is provided about possible differences in pigmentation or yields between young and old donors. Only on lines, 417-419 something is mentioned about the possible difference between samples. Moreover, authors mentioned that several samples were pooled and it seems that these pools (young versus old) were used in all the experiments, and the N reflects the number of replicates being statistical analysis done using the replicates. If so, this is incorrect and different pools need to be analysed if not individual samples.
  2. The number of granules used seems to vary from experiment to experiment, without any explanation of the reason for this variation.
  3. The choice of the combination of the antioxidants, as well as the concentration, also need to be explained.
  4. The necessity to pre-load the granules needs to be explained. The experiments should be repeated by pre-exposing the cells to the same dose of antioxidants to better represent what might in the future represent a preventive treatment.

Author Response

Response to reviewer #1 comments

We thank the reviewer  for thorough review of our manuscript and their thoughtful comments that helped us to prepare an improved version of  the revised manuscript. Point-by point replies to all comments are as follows:

The manuscript by Olchawa et al demonstrates that the antioxidants zeaxanthin and alpha-tocopherol partially prevented the harmful effects caused by melanolipofuscin granules isolated from human eyes. Generally, the manuscript is clear and the data is well presented. However, several aspects need to be clarified before this manuscript can be accepted for publication.

Minor:

The manuscript needs to be carefully edited to correct some grammar and spelling errors. Hereby some examples are presented:
Line 86: “enrichement of pigmet” should be “enrichment of pigments”
Line 140: “APRE-19” should be “ARPE-19”
Line 221: “supplenemtation” should be “supplementation”
Line 401: “signals” should be “signals”
Line 36: “is the outermost part of the eye retina”, the sentence should be rephrased
Line 41: “Recently,” refers to work published in 2017 and therefore not very recent

These grammar and spelling errors have been corrected or changed as suggested by the Reviewer (p.3, l. 96; p.4, l. 155; p.6, l. 239; p.12, l. 422; p.1, l. 36; p.1, l. 41-42).

Major:

The material and methods section needs to be re-written to allow others to better understand what was done and how was it done. This section has too many references what reduces the readability of the paper.  Line 66: Materials sections might be removed, and the materials used mentioned in the different following sections.

We wanted to provide the necessary information about the material and methods used by us, which is normally required for independent reproduction of the study main results. In order to make the description as concise as possible, we referred to papers that provide detailed description of specific methods and procedures. Following the reviewer’s suggestion, some additional information was added to material and methods section (p. 2-6).

Line 86: Isolation of the granules: The information of how RPE was isolated, from how many eyes, the characteristics of the donors and the post-mortem time from the death until the isolation is lacking and needs to be added to the manuscript.
Line 94: The authors need to add details of how the pellets were obtained.

Following the reviewer’s suggestion, we have included additional information about the protocol of RPE isolation and handling (p.3, l. 98-101), and about the number of eyes from younger and older donors (p.3, l. 103) as well as more detailed description how the pellets were (p.3, l. 104-106).

Line 99 and 104: It is quite difficult to understand how the granules were used; the authors should add a scheme explaining it.

A more detailed description of the protocol of the granule enrichment with antioxidants is now included (p. 3, l. 111-117; l. 118-122).

Line 100: “injected” should be “added”.

We changed “injected” to “added” as suggested by the reviewer (p.3, l. 112).

Line 110: The information about the type of dishes used is not informative and it should be removed. However, it should be clear the culture medium used in each experiment.

The type of dishes determines the number of cells and the amount of material used in specific experiments, therefore this information is viewed as relevant. The information about the culture medium used was present in our original submission (p. 3, l .126-127).

Line 139: How the POS were isolated needs to be clearly stated.

Following to reviewer’s suggestion we now added additional information about the protocol of POS isolation (p.4, l. 154, l. 157-161).

One of the main issues of this work is related to sampling. No information is provided about possible differences in pigmentation or yields between young and old donors. Only on lines, 417-419 something is mentioned about the possible difference between samples.

We have added information about the differences between the amount of isolated granules from younger and older donors (p. 6, l. 250-251). A short comment about visually observed differences in pigmentation of MLF granules between both age group was also included (p.6, l. 252-253).

 Moreover, authors mentioned that several samples were pooled and it seems that these pools (young versus old) were used in all the experiments, and the N reflects the number of replicates being statistical analysis done using the replicates. If so, this is incorrect and different pools need to be analysed if not individual samples.

To carry out the planned experiments, we needed significant amount of the human material that for obvious reasons, is of limited availability. The isolation of melanolipofuscin granules was done three times for each age group of human donors. For each isolation, 30-35 eyes per group was used  with the total number of eyes per each donor group being around 100. In the described experiments, three different pools of granules for each donors group were used. N is the number of independent experiments that were carried out with three different pools of MLF granules. It was impossible to analyze samples from individual donors because the amount of the obtained pigment granules was simply inadequate. I addition, experiments on pooled samples minimized significant differences in individual pigmentation of RPE cells, which could affect the age-related differences. Our statistical analysis showed the significance of differences between effect mediated by MLF granules isolated from younger and older donors and the modulatory effect of antioxidants.

The number of granules used seems to vary from experiment to experiment, without any explanation of the reason for this variation.

The number of granules used in our cellular studies varied between different experiments because in vitro studies required much more material, whereas photoreactivity experiments needed much less material.

The choice of the combination of the antioxidants, as well as the concentration, also need to be explained.

The choice of the combination of antioxidants used in our study is explained on p. 13, l. 485-486; p.14, l. 487 and the material and methods section contain the information about the final antioxidant concentration (p.3, l. 115 and l. 120). Additional information about the selected antioxidants is now included in the amended Introduction  (p. 2, l. 59-69).

The necessity to pre-load the granules needs to be explained. The experiments should be repeated by pre-exposing the cells to the same dose of antioxidants to better represent what might in the future represent a preventive treatment.

In this study we  focused on analyzing the modulatory effects of selected natural antioxidants in melanolipofuscin-dependent phototoxicity. Considering that melanolipofuscin is present in the RPE as distinct pigment granules about 1 micron in size, the corresponding effects are likely to be site-specific. That is why we used melanolipofuscin granules enriched with the selected antioxidants rather than cells pre-treated with the antioxidants. Experiments, in which cells will be pre-exposed to antioxidant will be carried out in the future. 

Reviewer 2 Report

The present study from Olchawa et al.  is of undoubted interest, although maybe a little complicated. Authors investigated how Melanolipofuscin (MLF) granules isolated from human retinal pigment epithelium, act on Human ARPE-19 cells engulfing them. The idea is nice, as it allowed to use human MLF granules in a cell culture setting.The interest of the manuscript lays in its use of human MLF granules isolated from donors of different age-span. The difference among MLF of different ages is also very interesting.

However, it seems that multiple aims were seeked for: although the declared aim of the study was to determine the phototoxic potential of melanolipofuscin granules from younger and older human donors to, and to examine if natural antioxidants  could modify  photogeneration of reactive oxygen species, which appears not surprisingly to be true there seem to be multiple aims, that render the reading a little difficult. melanolipofuscin photo-generates superoxide anion inducing oxidation of cellular proteins and disturbing the organization of cytoskeleton of RPE cells, and inhibit their phagocytic  activity.

Anyway,  the study correctly designed, analyses were performed in an appropriate way and with adequate technical standards. Methods and reagents are described with enough detail. Presentation of the results is correct, and Literature was appropriately cited.

In summary the paper showed that MLF exhibits significant aerobic photoreactivity increasing with aging, that can be counteracted by zeaxanthin and  a−tocopherol pretreatment of granules for 24 h, that the Authors ascribe to the photochemical properties of lipofuscin and melanin components of melanolipofuscin, which is debatable.
There is a possible bias in that the sample , already possibly degraded due to its provenience can differently suffer from the isolation manipulations due to its difference a priori (donor age difference and the different composition in light-sensitive substances).

The oxygen consumption cannot simply mean that the cell is going to be more oxidized, as aerobic metabolism also allows the cell to have sufficient chemical energy to operate detoxification of reactive oxygen species.

The human ARPE-19 cell model can recapitulate features of RPE pathology associated with blinding diseases in which the RPE is severely impaired by oxidative stress.

Notably it has been reported that blue light causes oxidative stress that is stronger in the OS than in the IS where the mitochondria are located. doi: 10.1371/journal.pone.0071570.

the Authors completely disregarded the literature  that demonstrates the rod OS are the site of an ectopic oxidative phosphorylation   that is severely impaired by blue light irradiation (doi: 10.1016/j.biochi.2016.03.016.). In doing this, they failed to frame the topic in the perspective  that their experimental  model allows to presume that the ultimate source of oxidative stress in the outer retina, responsible for the onset of most of retinal neurodegenerative diseases can come from the rod outer segments, as hsown indirectly by the fact that with age the degree of melanolipofuscins increases ( the RPE would have engulfed continually more and more rod disks  gradually more oxidized as o the oxidative stress that takes place therein was increasing with age).

The ectopic respiratory chain would be a primary unshielded source of Reactive Oxygen Species that may be one cause of the rod-driven macular diseases in humans. An extra-mitochondrial oxidative phosphorylation was reported also in many cellular membranes (see doi: 10.1016/j.bbabio.2008.08.003.; Shimizu N, et al. Am J Physiol Heart Circ Physiol. 2007 Sep;293(3):H1646-53), and also in platelets (doi: 10.1111/boc.201700025.) and extracellular vesicles (DOI: 10.1080/14789450.2018.1528149).

On the other hand, it has been known for a long time that oxygen is absorbed mainly at the level of the photoreceptors (Invest Ophthalmol Vis Sci 31:1029-1034,1990) and that the respiratory chain is the main producer of Reactive Oxygen Species. Recent evidence shows that the primary retinal damage from oxidative stress production occurs to rod outer segments and not only in the inner segment containing the mitochondria, but also in the Outer limb (see  doi: 10.5318/wjo.v4.i3.29).

The rod outer segment can become a strong free radical producer when the ectopic redox chain or the membranes it is embedded in become impaired, in order to proprly set up the scientific issue of POS degeneration. In fact, the photoreceptor apoptosis starts from the OS.

this can be a work subtly approaching the issue of the origin of oxidative stress in the outer retina, up to now attributed to the RPE itself. Instead, this Reviewer is convinced that the RPE is an innocent bystander and is not, as claimed for years in literature, the origin of the outer retinal oxidative stress.

Therefore Authors should consider that their experimental model showed evidence that the oxidized MLF can damage the RPE and which bears the potential to overturn the current vision of the pathogenesis of AMD, diabetic retinopathy and other neurodegenerations, with enormous repercussions in the field. I suggest taking into consideration these emerging data.

Author Response

Response to reviewer #2 comments

We thank the reviewer  for thorough review of our manuscript and their thoughtful comments that helped us to prepare an improved version of  the revised manuscript. Point-by point replies to all comments are as follows:

The present study from Olchawa et al.  is of undoubted interest, although maybe a little complicated. Authors investigated how Melanolipofuscin (MLF) granules isolated from human retinal pigment epithelium, act on Human ARPE-19 cells engulfing them. The idea is nice, as it allowed to use human MLF granules in a cell culture setting. The interest of the manuscript lays in its use of human MLF granules isolated from donors of different age-span. The difference among MLF of different ages is also very interesting.
However, it seems that multiple aims were seeked for: although the declared aim of the study was to determine the phototoxic potential of melanolipofuscin granules from younger and older human donors to, and to examine if natural antioxidants  could modify  photogeneration of reactive oxygen species, which appears not surprisingly to be true there seem to be multiple aims, that render the reading a little difficult. melanolipofuscin photo-generates superoxide anion inducing oxidation of cellular proteins and disturbing the organization of cytoskeleton of RPE cells, and inhibit their phagocytic  activity.

The reported by us melanolipofuscin-mediated photooxidation of cellular proteins and disruption of the cell cytoskeleton, most likely responsible for the observed inhibition of the specific phagocytic activity of ARPE-19 cells, is consistent with our main  research aim, which was the analysis of phototoxic potential of human RPE melanolipofuscin. As stressed in the manuscript, our goal was to examine sub-lethal effects mediated by melanolipofuscin that in vivo might be responsible for deterioration of the main RPE function - its phagocytosis of photoreceptor outer segments.

Anyway,  the study correctly designed, analyses were performed in an appropriate way and with adequate technical standards. Methods and reagents are described with enough detail. Presentation of the results is correct, and Literature was appropriately cited.
In summary the paper showed that MLF exhibits significant aerobic photoreactivity increasing with aging, that can be counteracted by zeaxanthin and  a−tocopherol pretreatment of granules for 24 h, that the Authors ascribe to the photochemical properties of lipofuscin and melanin components of melanolipofuscin, which is debatable.

Although we respect the reviewer’s reservation about the role of melanin and lipofuscin components in the resultant photoreactivity and phototoxicity of the examined melanolipofuscin granules, the photochemical and photophysical properties of RPE pigment granules we have analyzed extensively over the years, are fully consistent with the conclusions of our study.

There is a possible bias in that the sample, already possibly degraded due to its provenience can differently suffer from the isolation manipulations due to its difference a priori (donor age difference and the different composition in light-sensitive substances).

The reviewer’s considerations about possible differential degradation of melanolipofuscin granules due to isolation procedure and different donors’ age might be legitimate; however, it is not supported by any study we are aware of. Indeed, our experience in dealing with pigment granules isolated from RPE, convinces us that the procedures we employed to isolate and handle melanolipofuscin granules was fully adequate to achieve the main goals of the study. Moreover our control experiments indicated that any compositional changes of lipofuscin granules or melanosomes associated with senescence, had no effect on the ability of RPE cells to phagocytize the granules.

The oxygen consumption cannot simply mean that the cell is going to be more oxidized, as aerobic metabolism also allows the cell to have sufficient chemical energy to operate detoxification of reactive oxygen species.

There seems to be a misunderstanding of the meaning of the observed photoconsumption of oxygen reported in this study. While there is no doubt that due to aerobic metabolisms, cells consume oxygen,  in our experimental set up we monitored consumption of oxygen induced by irradiation with blue light of cells containing melanolipofuscin granules with and without antioxidants. In the dark, under the  conditions used, there was no measurable oxygen consumption and, likewise, almost no oxygen consumption was detected when cells without melanolipofuscin were irradiated with the same light.

The human ARPE-19 cell model can recapitulate features of RPE pathology associated with blinding diseases in which the RPE is severely impaired by oxidative stress.
Notably it has been reported that blue light causes oxidative stress that is stronger in the OS than in the IS where the mitochondria are located. doi: 10.1371/journal.pone.0071570.
the Authors completely disregarded the literature  that demonstrates the rod OS are the site of an ectopic oxidative phosphorylation   that is severely impaired by blue light irradiation (doi: 10.1016/j.biochi.2016.03.016.). In doing this, they failed to frame the topic in the perspective  that their experimental  model allows to presume that the ultimate source of oxidative stress in the outer retina, responsible for the onset of most of retinal neurodegenerative diseases can come from the rod outer segments, as hsown indirectly by the fact that with age the degree of melanolipofuscins increases ( the RPE would have engulfed
continually more and more rod disks  gradually more oxidized as o the oxidative stress that takes place therein was increasing with age
The ectopic respiratory chain would be a primary unshielded source of Reactive Oxygen Species that may be one cause of the rod-driven macular diseases in humans. An extra-mitochondrial oxidative phosphorylation was reported also in many cellular membranes (see doi: 10.1016/j.bbabio.2008.08.003.; Shimizu N, et al. Am J Physiol Heart Circ Physiol. 2007 Sep;293(3):H1646-53), and also in platelets (doi: 10.1111/boc.201700025.) and extracellular vesicles (DOI: 10.1080/14789450.2018.1528149).

Although we agree with the reviewer that “ARPE-19 cell model can NOT FULLY recapitulate features of RPE pathology associated with blinding diseases”, this cell line is a convenient model for studying various deleterious effects of oxidative stress. The limitations of using the ARPE-19 model are discussed in the amended manuscript (p. 16, l. 570-582). In this study, we have focused on melanolipofuscin as an endogenous photosensitizer that may contribute to light-induced oxidative stress in the outer retina. In our previous studies, we analyzed the potential role of lipofuscin, aged melanin and oxidized lipids in such processes. Although we appreciate the reviewer’s suggestion that ectopic  respiratory chain could be an important source of reactive oxygen species, our study deals only with photosensitized oxidative stress mediated by melanolipofuscin.

On the other hand, it has been known for a long time that oxygen is absorbed mainly at the level of the photoreceptors (Invest Ophthalmol Vis Sci 31:1029-1034,1990) and that the respiratory chain is the main producer of Reactive Oxygen Species. Recent evidence shows that the primary retinal damage from oxidative stress production occurs to rod outer segments and not only in the inner segment containing the mitochondria, but also in the Outer limb (see  doi: 10.5318/wjo.v4.i3.29).

The rod outer segment can become a strong free radical producer when the ectopic redox chain or the membranes it is embedded in become impaired, in order to proprly set up the scientific issue of POS degeneration. In fact, the photoreceptor apoptosis starts from the OS.
this can be a work subtly approaching the issue of the origin of oxidative stress in the outer retina, up to now attributed to the RPE itself. Instead, this Reviewer is convinced that the RPE is an innocent bystander and is not, as claimed for years in literature, the origin of the outer retinal oxidative stress.

We respect the reviewer’s belief that RPE is “an innocent bystander and is not, as claimed for years in literature, the origin of the outer retinal oxidative stress”; however,  unless proven otherwise, we agree with many researchers  that the available experimental data reasonably support the key role of RPE in the onset of AMD. As for the potential role of photoreceptor outer segments in light-induced oxidative stress, we believe that POS, particularly after extensive photobleaching of the visual pigments, could also play a role. Indeed we have addressed this issue in a recently published paper “In vitro phototoxicity of rhodopsin photobleaching products in the retinal pigment epithelium  (RPE)” (M. Olchawa et al., Free Radical Research 2019).

Therefore Authors should consider that their experimental model showed evidence that the oxidized MLF can damage the RPE and which bears the potential to overturn the current vision of the pathogenesis of AMD, diabetic retinopathy and other neurodegenerations, with enormous repercussions in the field. I suggest taking into consideration these emerging data.

It is widely believed that oxidative stress plays an important role in the development of age-related macular degeneration. Although the main source of reactive oxygen species responsible for oxidative stress of the outer retina and the cause of failure of the protective antioxidant systems maybe disputable, our study indicates that photoreactivity of the accumulating with age melanolipofuscin could be involve in chronic phototoxicity of the RPE.

Reviewer 3 Report

This is a fine paper describing the phototoxicity mediated by internalized melanolipofuscin granules with and without supplementation with zeaxanthin and alpha-tocopherol. 

Overall, the methodology and data are described in sufficient detail. 

This reviewer's primary concern is the use of ARPE-19 cells as a model for AMD. It is well established in the field that ARPE-19 cells do not mimic key physiological hallmarks of primary RPE cells. These include pigmentation, permeability,  and proliferation. Furthermore, ARPE-19 cells possess a differential endogenous antioxidant response. 

Consequently, the results described in this manuscript may not be as far reaching as anticipated. This reviewer feels strongly that the limitations associated with ARPE-19 cells need to discussed on the poser. 

Author Response

Response to reviewer #3 comments

We thank the reviewer  for thorough review of our manuscript and their thoughtful comments that helped us to prepare an improved version of  the revised manuscript. Point-by point replies to all comments are as follows:

This is a fine paper describing the phototoxicity mediated by internalized melanolipofuscin granules with and without supplementation with zeaxanthin and alpha-tocopherol.
Overall, the methodology and data are described in sufficient detail.

This reviewer's primary concern is the use of ARPE-19 cells as a model for AMD. It is well established in the field that ARPE-19 cells do not mimic key physiological hallmarks of primary RPE cells. These include pigmentation, permeability,  and proliferation. Furthermore, ARPE-19 cells possess a differential endogenous antioxidant response.
Consequently, the results described in this manuscript may not be as far reaching as anticipated. This reviewer feels strongly that the limitations associated with ARPE-19 cells need to discussed on the poser.

We are aware about the limitation of ARPE-19 cells as a model for AMD. Unfortunately, there are no other in vitro models without such limitations. In spite of many drawbacks, ARPE-19 cells form morphologically and functionally polarized epithelial monolayers similar to those of RPE in vivo, and we and others, have shown that this stable cell line is a convenient in vitro model for studying how oxidative stress affects various biological functions of the cells, including their ability to phagocytose POS discs. In addition, we demonstrated that under appropriate culture conditions ARPE-19 cells express αvβ5 integrin and MerTK receptor proteins, which are essential for efficient phagocytosis of POS discs (Olchawa et al,2013).  It must be stressed that alternative RPE cultures have also significant problems. Thus other widely-used lines are from rodents (e.g., the rat RPE-J line), which exhibit species differences. Although cultures can also be propagated from the RPE of adult human donor eyes, they are highly variable making outcomes from such cultures less generalizable (Burke J. et al., 1996, Burke J. et al., 2008). The most representative RPE cultures may be those from human fetal eyes, but these cells come from an early stage of development and may not resemble the tissue from the adult eye, and they are  difficult to obtain and to propagate. The limitations associated with ARPE-19 cells are discussed by us in the revised manuscript (p. 16, l. 570-582). In our study, we addressed two specific issues: 1. Does photoreactivity (and potential phototoxicity) of melanolipofuscin granules isolated from human RPE change with age of the donors and if can be modulated by natural antioxidants, 2. Does photic stress mediated by melanolipofuscin granules inhibit specific phagocytosis of ARPE-19 cells. These issues have been examined by us in carefully designed experiments yielding results that, at least partially, could be extrapolated to other cells and even to RPE in vivo.

Reviewer 4 Report

In this study, the authors have investigated whether melanolipofuscin granules obtained from younger and older human donors could photo-generate ROS and antioxidants such as zeaxanthin and a-tocopherol could change the photoreactive and phototoxic potential of this age pigment. This manuscript seems to have been written very well, and the results are very interesting and well developed. This reviewer does not feel there is much to revise, but there are two points that can be improved. (1) The introduction section has not been clearly introduced and is described too short, maybe the authors could supply more. For example, one of the important parts of the paper is the antioxidants including zeaxanthin, and a-tocopherol, but none of them is mentioned in the introduction. (2) If the authors could make the language more native format, this manuscript will be more attractive for readers.

Minor point:

Figure 3b, MLF19-20 should be MLF18-29

Author Response

Response to reviewer #4 comments

We thank the reviewer  for thorough review of our manuscript and their thoughtful comments that helped us to prepare an improved version of  the revised manuscript. Point-by point replies to all comments are as follows:

In this study, the authors have investigated whether melanolipofuscin granules obtained from younger and older human donors could photo-generate ROS and antioxidants such as zeaxanthin and a-tocopherol could change the photoreactive and phototoxic potential of this age pigment. This manuscript seems to have been written very well, and the results are very interesting and well developed. This reviewer does not feel there is much to revise, but there are two points that can be improved. (1) The introduction section has not been clearly introduced and is described too short, maybe the authors could supply more. For example, one of the important parts of the paper is the antioxidants including zeaxanthin, and a-tocopherol, but none of them is mentioned in the introduction. (2) If the authors could make the language more native format, this manuscript will be more attractive for readers.

Following the reviewer’s suggestions, the introduction has been expanded by addition of a short description of the antioxidants used in this study (p. 2, l. 59-69).

Minor point:

Figure 3b, MLF19-20 should be MLF18-29

The indicated typo has been corrected (p.8).

Reviewer 5 Report

Authors investigated the effect of antioxidants on photoreactivity and phototoxic potential of retinal pigment epithelium (RPE) melanolipofuscin granules from human donors of different age. The manuscript is well written.

How many donors were pooled into the two age groups? Around 10? 100? 1000?
The number is necessary for generalizing the conclusion.

Author Response

Response to reviewer #5 comments

We thank the reviewer  for thorough review of our manuscript and their thoughtful comments that helped us to prepare an improved version of  the revised manuscript. Point-by point replies to all comments are as follows:

Authors investigated the effect of antioxidants on photoreactivity and phototoxic potential of retinal pigment epithelium (RPE) melanolipofuscin granules from human donors of different age. The manuscript is well written.
How many donors were pooled into the two age groups? Around 10? 100? 1000?
The number is necessary for generalizing the conclusion.

Melanolipofuscin granules isolation was done three times for each donor age group and for each isolation we used 30-35 eyes per age-group (total number of eyes per each donor group was around 100).

Round 2

Reviewer 1 Report

The manuscript still has some typos example: Line 96: "pigmetns". The authors failed in providing detailed information in the material and section, for example, Line 105 and 121, please indicate for how long and what was the speed used in the centrifugation. line 570-582: I advise to ask the authors to revise the paragraph. The tone and language are not appropriate for a scientific publication.

Author Response

Again, we thank the reviewer for helping to improve our manuscript. Following the reviewer's suggestion, we have provided detailed information in the material and methods section. We have also amended the indicated by reviewer paragraph.

Reviewer 3 Report

The authors have adequately addressed the comments provided in response to the original submission. Congratulations on this fine work. 

Author Response

We appreciate the reviewers comments.